# Combining the Potent Reducing Properties of Pecan Nutshell with a Solvent-Free Mechanochemical Approach for Synthesizing High Ag^0^ Content-Silver Nanoparticles: An Eco-Friendly Route to an Efficient Multifunctional Photocatalytic, Antibacterial, and Antioxidant Material

**DOI:** 10.3390/nano13050821

**Published:** 2023-02-23

**Authors:** Rita Argenziano, Sarai Agustin-Salazar, Andrea Panaro, Anna Calarco, Anna Di Salle, Paolo Aprea, Pierfrancesco Cerruti, Lucia Panzella, Alessandra Napolitano

**Affiliations:** 1Department of Chemical Sciences, University of Naples “Federico II”, Via Cintia 4, I-80126 Naples, Italy; 2Institute for Polymers, Composites and Biomaterials (IPCB-CNR), Via Campi Flegrei 34, I-80078 Pozzuoli, Italy; 3Research Institute on Terrestrial Ecosystems (IRET-CNR), Via P. Castellino 111, I-80131 Naples, Italy; 4Department of Chemical, Materials and Industrial Production Engineering, University of Naples “Federico II”, Piazzale V. Tecchio 80, I-80125 Naples, Italy

**Keywords:** silver nanoparticles, green chemistry, mechanochemistry, waste valorization, condensed tannins

## Abstract

A straightforward, low-cost, and scalable solid-state mechanochemical protocol for the synthesis of silver nanoparticles (AgNP) based on the use of the highly reducing agri-food by-product pecan nutshell (PNS) is reported herein. Under optimized conditions (180 min, 800 rpm, PNS/AgNO_3_ ratio = 55/45 *w*/*w*), a complete reduction in silver ions was achieved, leading to a material containing ca. 36% *w*/*w* Ag^0^ (X-ray diffraction analysis). Dynamic light scattering and microscopic analysis showed a uniform size distribution (15–35 nm average diameter) of the spherical AgNP. The 2,2-Diphenyl-1-picrylhydrazyl (DPPH) assay revealed lower—although still absolutely high (EC_50_ = 5.8 ± 0.5 mg/mL)—antioxidant properties for PNS for the further incorporation of AgNP, supporting the efficient reduction of Ag^+^ ions by PNS phenolic compounds. Photocatalytic experiments indicated that AgNP-PNS (0.4 mg/mL) was able to induce the >90% degradation of methylene blue after 120 min visible light irradiation, with good recycling stability. Finally, AgNP-PNS demonstrated high biocompatibility and significantly light-enhanced growth inhibition properties against *Pseudomonas aeruginosa* and *Streptococcus mutans* at concentrations as low as 250 μg/mL, also eliciting an antibiofilm effect at 1000 μg/mL. Overall, the adopted approach allowed to reuse a cheap and abundant agri-food by-product and required no toxic or noxious chemicals, making AgNP-PNS a sustainable and easy-to-access multifunctional material.

## 1. Introduction

Silver nanoparticles (AgNP) are currently finding wide applications not only in electronics, photonics, and catalysis [1,2], but also in health-related fields, such as antitumor, bactericidal, and antifungal agents [3,4,5,6]. The AgNP preparation method is rather critical, since their cytotoxicity strongly depends on their shape and size, and the silver ion reduction is often incomplete, leading to mixtures of Ag^0^ and Ag^+^ nanoparticles [7,8,9]. In addition, to make the AgNP preparation more sustainable in a green chemistry context, protocols envisaging clean synthetic approaches and/or green reducing agents are currently the focus of increasing interest [10,11,12,13,14,15,16].

Among the first sustainable and versatile synthetic methodologies to have attracted increasing attention is mechanochemistry. Mechanochemical reactions are induced by the absorption of mechanical energy, and have been widely explored for the synthesis of both inorganic materials and valuable organic compounds, including AgNP, with the huge advantage of avoiding the use of solvents [17,18,19,20,21,22,23]. Furthermore, the mechanochemical synthesis generally does not require any post-reaction treatment to isolate the product. The latter can be directly stored for a long time as a dry powder, while NP prepared by solvent-assisted methods tend to aggregate upon storage.

As far as sustainable reducing and stabilizing agents are concerned, agri-food wastes and by-products rich in lignins, tannins, and other phenolic compounds endowed with potent antioxidant properties are increasingly being employed for the synthesis of AgNP with reduced toxicity and improved effectiveness [24,25]. Remarkable and recent examples are viticultural wastes [26,27], cereal husks [28], pecan leaves [29], and spent coffee grounds [30]. We recently reported the very efficient antioxidant properties of pecan nut shell (PNS) [31] and its hydroalcoholic extract [32], which are characterized by lignin and condensed tannins as the main phenolic components [31,32,33]. In particular, in a comparison between several plant-derived biowastes using widely used assays, PNS showed the highest antioxidant activity [31].

Several uses of this largely produced agri-food by-product (0.12 million tons per year in U.S.A. only) [34] and the compounds thereof have indeed been proposed, ranging from food packaging [32,35,36] and polymer functionalization and stabilization [37,38], to water decontamination [34,39] and a fungi or bacteria carbon source for bioactives production [40]. Actually, the use of PNS extracts for the preparation of antimicrobial AgNP has recently been reported [41]. However, the possibility of using solid PNS as such, combined with a solvent-free mechanochemical approach, for the implementation of multifunctional devices with antioxidant, antimicrobial, and photocatalytic activities, has never been explored.

In light of these observations, and considering the remarkable antioxidant activity of PNS, its potential use for the preparation of AgNP is reported herein. Two different approaches were initially compared, namely the conventional wet-chemical and mechanochemical synthesis routes, and the latter turned out to be more efficient in terms of Ag^+^ reduction. Therefore, the samples obtained by the optimized mechanochemical protocol, hereafter referred to as AgNP-PNS, were characterized by X-ray diffraction analysis (XRD), attenuated total reflection (ATR)-FTIR spectroscopy, dynamic light scattering (DLS), and transmission (TEM) and scanning (SEM) electron microscopy. Finally, the photocatalytic, antimicrobial, and antioxidant properties of AgNP-PNS were evaluated.

## 2. Materials and Methods

### 2.1. Materials

PNS were supplied by Asociación Productora de Nuez S.P.R de R.I. (Hermosillo, Mexico). All reagents and solvents were of analytical grade (Sigma-Aldrich, Steinheim, Germany). *Pseudomonas aeruginosa* PAO1 (ATCC^®^ BAA-47) and *Streptococcus mutans* (ATCC^®^ 25175) were supplied by the American Type Culture Collection (ATCC) via a local distributor (LGC Standards S.r.l, Sesto San Giovanni, Italy). Luria–Bertani (LB) agar, nutrient agar and BacTiter-Glo™ Microbial Cell Viability Assay Reagent were purchased from Promega (Milan, Italy).

### 2.2. PNS Milling and Sieving

PNS were dried in an oven at 50 °C for 16 h and finely minced in a grinding mill (Retsch ZM1 GmbH & Co., KG 5657, Haan, Germany for 15 min. Fifteen grams of the sample was then introduced into a vibratory sieve shaker (Retsch AS 200 basic B, Retsch GmbH Retsch-Allee 1-5 42781 Haan, Germany) to obtain a material with homogeneous particle size distribution (75–125 μm).

### 2.3. Preparation of AgNP by the Wet-Chemical Approach

PNS (200 mg) was added to 20 mL of a AgNO_3_ 10 mM water solution, and the suspension was stirred at room temperature for 24 h. The mixture was then centrifuged at 7000 rpm for 30 min, and the precipitate was washed three times with water and lyophilized to afford 150 mg of a black powder. For comparison purposes, the same experimental protocol was performed using grape pomace or pomegranate peel and seeds as a reducing agent.

### 2.4. Preparation of AgNP by the Mechanochemical Approach

PNS was dried in a vacuum oven at 60 °C for 16 h. A total of 140 mg of PNS was then milled with 25 mg of AgNO_3_ (15% *w*/*w*) using a vibratory ball mill (WIG-L-BUG amalgamator model 3110-3A, Lyons, Ill., USA at 60 Hz for 90 min. In a scaled-up procedure, the reaction was performed in a planetary ball mill (Retsch PM100, Haan, Germany) at 800 rpm at different concentrations, all of them are reported in Table 1.

### 2.5. Characterization of AgNP-PNS

AgNP-PNS was characterized by XRD, SEM, TEM, DLS, and ATR-FTIR spectroscopy. XRD analysis was performed by a Malvern PANalytical X’Pert Pro diffractometer with a PIXCel 1D detector, using CuKα radiation (Malvern, Worcestershire, UK). Powder spectra were collected in a 2θ range of 5–80°, with steps of 0.013° and a counting time of 20 s per step. The crystalline phases in the samples were identified with the PANalytical HighScore software v5.1 (Malvern, Worcestershire, UK), equipped with the ICDD PDF 2 database. The RIR/Rietveld method [42] was used to measure the amount of AgNP in the samples, using a weighted amount (10% *w*/*w*) of corundum (NIST Standard Reference Material 676a) as standard in each sample. Quantitative phase analysis was performed using the GSAS-II software [43] to obtain the relative mass percentage of corundum and Ag^0^. The absolute weight percentage of each phase was then obtained by rescaling the results according to the actual amount of corundum. Each measure was performed in triplicate. The structural information of each phase was acquired from the Crystallography Open Database [44].

ATR-FTIR spectra were recorded on the solid samples using a Nicolet 5700 Thermo Fisher Scientific instrument (Verona Rd. Madison, WI, USA). Spectra were recorded in the 4000–450 cm^−1^ range (resolution of 4 cm^−1^), as an average of 32 scans.

For SEM, TEM, and DLS characterization, 10 mg of the sample were suspended in 3 mL of distilled water. The mixture was sonicated in a ultrasonic bath (Branson B-2200 E1, Branson Ultrasonics corporation, Eagle Rd. Danbury, Connecticut, USA) for 30 min, and then allowed to settle for another 30 min. The obtained supernatant was analyzed by TEM (FEI Tecnai G2 Spirit Twin device LaB6 source, Eindhoven, The Netherlands) and TEM images were taken with a FEI Eagle 4 k CCD camera, SEM (FEI Quanta 200 FEG, 10–30 kV acceleration voltage, secondary electron detector, Eindhoven, The Netherlands), and DLS at 25 °C, with a wavelength of 633 nm and the detection of backscattered light at an angle of 173°, (Malvern NanoSizer ZS, Malvern instruments, (Malvern, Worcestershire, UK).

### 2.6. Antioxidant Properties Evaluation

*2,2-Diphenyl-1-picrylhydrazyl (DPPH) assay.* Each sample (0.025–7.5 mg/mL) was added to a 200 μM DPPH solution in ethanol, [30,45] and stirred for 10 min at room temperature. The absorbance at 515 nm was then measured with a Jasco V730 UV–Vis spectrophotometer. The experiments were run at least in triplicate. *Ferric reducing/antioxidant power (FRAP) assay.* Each sample (0.03–0.15 mg/mL) was added to a solution composed of 20 mM FeCl_3_, 10 mM 2,4,6-tris(2-pirydyl)-s-triazine, and 0.3 M acetate buffer (pH 3.6) in a 1:1:10 *v*/*v*/*v* ratio [30,46]. After stirring for 10 min at room temperature, the absorbance at 593 nm was measured. Experiments were run in triplicate.

### 2.7. Photocatalytic Properties Evaluation

Each sample (4 mg) was added to 10 mL of a methylene blue (MB) solution (30 mg/mL) in water in a 60 mm cell-culture dish. The suspension was irradiated for 120 min using a solar simulator (Thermo Oriel 66902 model, Stratford, CT, USA) at 60 W, with a UV-cutoff filter (λ > 400 nm). Two milliliters of the mixture was withdrawn at 30 min intervals, analyzed by a UV–Vis spectrophotometer Jasco V730 (JASCO Europe, Cremella, Lecco, Italy), and carefully replaced in the cell culture dish. Control experiments were performed: (i) in the absence of the sample; (ii) in the dark; and (iii) in the dark only for the first 30 min (in the case of the AgNP-PNS sample). Each experiment was run in triplicate. When required, at the end of the experiment, AgNP-PNS was recovered by centrifugation, washing, and lyophilization, and reused in the photodegradation experiment as above.

### 2.8. Biocompatibility Evaluation

Human dermal fibroblasts (HDFs) and human epidermal keratinocytes (HEKs) were obtained from the American Type Culture Collection (ATCC, Manassas, VA, USA) and cultured in Dulbecco’s modified Eagle’s medium (DMEM) and Ham’s F12 Nutrient Mixture (1:3) supplemented with 10% fetal bovine serum (FBS), penicillin (100 mg/mL), and streptomycin (100 U/mL) at 37 °C in a 5% CO_2_ atmosphere. To assess the cytotoxic effect of AgNP-PNS, the conventional CCK-8 assay (Sigma-Aldrich, Milan, Italy) was used as reported by Conte et al. [47]. Briefly, HDF and HEK cells were seeded on 96-well microplates and treated with AgNP-PNS (0–150 µg/mL) in a serum-free medium for 72 h. Cells incubated without AgNP-PNS were used as control. The optical density of formazan salt at 450 nm was measured using a Cytation 3 microplate reader (ASHI, Milan, Italy). Nanoparticles’ cytocompatibility was expressed as a percentage relative to the control and calculated by the following equation:$$\mathrm{Cytotoxicity}\;\left(\%\right)=\frac{A_{s}-A_{B}}{A_{c}-A_{B}}\times 100$$
where *A_S_* is the absorbance of the cells treated with AgNP-PNS, *A_B_* is the absorbance of background, and *A_C_* is the absorbance of the cells without AgNP-PNS.

Samples with viability percentages below 70% were considered cytotoxic according to the recommendations of ISO 10,993–5 [48].

### 2.9. Antibacterial Activity Evaluation

*P. aeruginosa* PAO1 (ATCC^®^ BAA-47) and *S. mutans* (ATCC^®^ 25175) were cultured according to the ATCC guidelines for 18 h on trypticase soy broth agar and trypticase soy yeast extract agar (Thermo Fisher Scientific, Waltham, MA, USA), respectively. One colony was then resuspended in liquid broth medium (5 mL) and incubated at 200 rpm and 37 °C overnight. The bacterial growth inhibition capability of AgNP-PNS was measured by measuring the optical density (OD) at 600 nm of bacterial suspensions cultured in the presence of the sample [49,50]. In particular, AgNP-PNS was placed at different concentrations (50, 100, 250, 500, and 1000 μg/mL) in a 96-well polystyrene plate in the presence of 200 μL liquid broth. Then, the bacteria were inoculated at 0.5 MacFarland standard turbidity (approximately 1.5 × 10^8^ CFU/mL) and the plates were incubated at 37 °C and 200 rpm in a microplate reader (Cytation 3, AHSI). Liquid medium broth without bacteria was used as negative control, while 200 μL of PAO1 (1.5 × 10^8^ CFU/mL) or *S. mutans* (1.5 × 10^8^ CFU/mL) were used as the positive controls. At scheduled times (6 h, 12 h, and 24 h), the OD at 600 nm was recorded. The experiments were performed in triplicate.

For the agar diffusion method, 100 μL of bacterial suspension containing approximately 1.5 × 10^8^ CFU/mL was evenly spread on the surface of a 10 mm × 150 mm solid broth medium and allowed to dry for approximately 10 min. Discs loaded with 250, 500, and 1000 μg/mL of AgNP-PNS or 1000 μg/mL of PNS were aseptically placed on the agar bacterial plate and incubated at 37 °C overnight either in dark or under light irradiation, using a desk lamp equipped with a 11 W, 806 lumen, E27 ID60 LED bulb (Jedi Lighting). A liquid medium broth without bacteria was used as the negative control. At the end of the incubation period, the diameters of the inhibition zones were measured [51]. The experiments were performed in triplicate.

### 2.10. Biofilm Growth Inhibition by AgNP-PNS

Biofilm was developed as described in [49] with some modifications. Briefly, different concentrations of AgNP-PNS (250, 500, and 1000 μg/mL) were placed in a 48-well polystyrene plate, in the presence of 750 μL PAO1 or *S. mutans* suspension (1 × 10^7^ CFU/mL). The cultures were statically incubated at 37 °C in a humid atmosphere for 6, 12, and 24 h, until a mature biofilm was obtained. The liquid medium broth without bacteria was employed as a negative control, while 750 μL of PAO1 (1 × 10^7^ CFU/mL) and *S. mutans* (1 × 10^7^ CFU/mL) were used as the positive controls.

The biofilm adhered to the surface was determined by the crystal violet (CV) assay [50]. Briefly, sterile phosphate-buffered saline (PBS) was used to gently wash each well. After 30 min air drying, a solution of 0.1% *w/v* CV was added to each well. Excess solution was removed after 30 min, and any extra stain was removed by washing with PBS. The stained biofilms were dissolved in 96% ethanol and quantification was performed by measuring the OD at 570 nm using a microplate reader (Cytation 3, AHSI). The experiments were performed in triplicate.

## 3. Results and Discussion

### 3.1. Silver Ion Reduction by PNS: Wet-Chemical vs. Mechanochemical Approach

The bioreductive synthesis of AgNP using plant derivatives usually leads to mixtures of AgCl and Agnanoparticles. The latter in particular are acknowledged as active antimicrobial agents as well as stable and efficient photocatalysts under visible light [51]. In consideration of the applications envisaged in the present paper, it was important to direct the synthesis towards AgNP rather than to AgClnanoparticles. To this aim, preliminary tests were carried out using a wet chemistry approach, employing water as solvent, and biomass from several agrifood byproducts, including PNS, grape pomace, and pomegranate peels and seeds, as bioreductive agents. Biomass particles with homogenous size distribution (75–125 μm) obtained by means of a vibratory sieve shaker (representing >80% *w*/*w* of the starting biomass) were employed for the preparation of AgNP.

For the wet-chemical approach, a previously developed experimental protocol was adopted [27] with slight modifications. Briefly, biomass (1% *w*/*v*) was taken under vigorous stirring in a 10 mM AgNO_3_ aqueous solution for 24 h after that the solid material was recovered in 75% *w*/*w* yield by centrifugation. It was found that PNS provided relatively high amounts of AgNP (3.7 ± 2% *w*/*w*), as confirmed by the presence of an intense diffraction peak at a 2θ = 38.3° without the formation of AgCl. The latter was otherwise found when the reaction was carried out in the presence of pomegranate peels and seeds (Appendix A). Therefore, PNS was selected as the biomass to be employed for AgNP synthesis in the subsequent mechanochemical approach. To this aim, a vibrational ball mill operating at 60 Hz was initially employed. The same PNS/AgNO_3_ ratio (85:15 *w*/*w*) as the wet-chemical protocol was used, and a milling time of 90 min was selected. The XRD analysis indicated the presence of pure Ag^0^ with a face-centered cubic structure, while no AgCl was found (Appendix A). The percentage of Ag^0^ was 8 ± 4% *w*/*w* (compared to a theoretical value of 9.5%), highlighting the superiority of the mechanochemical approach in terms of the efficacy of the Ag^+^ reduction process.

On this basis, further experiments were directed to the optimization of the experimental conditions for the mechanochemical preparation of AgNP. First, the effect of the PNS/AgNO_3_ ratio was evaluated, employing three different conditions, namely 85:15, 70:30, and 55:45 *w*/*w* PNS/AgNO_3_. The comparison of the respective XRD patterns (Appendix A) shows that the overall Ag^0^ yield was not substantially affected by the percentage of AgNO_3_, while the progressively reduced full-width at half maximum (FWHM) indicated an increase in the Ag^0^ crystallite size. Subsequently, in order to develop a larger-scale process, a planetary ball-mill was used, allowing to work on tens of grams of material. Different milling times and PNS/AgNO_3_ ratios were tested. The results of the XRD analysis (Appendix A) revealed that prolonging the milling time up to 180 min not only increased the amount of Ag^0^ in the sample (up to 7.5%), but also resulted in a lowered FWHM of the peak at 38.3°, suggesting an increased crystalline size. In particular, Appendix A shows that the FWHM of the peak decreases almost linearly with the treatment time.

An increase in the percentage content of Ag^0^ was also observed to be moving from a 85:15 *w*/*w* to a 55:45 *w*/*w* PNS/AgNO_3_ ratio, under which conditions a 36 ± 7% *w*/*w* Ag^0^ amount was calculated (Figure 1a), consistent with the expected theoretical value (about 30%). On this basis, the sample prepared by ball milling a 55:45 *w*/*w* PNS/AgNO_3_ dry mixture for 180 min in a planetary ball mill, hereafter referred to as AgNP-PNS, was selected as the optimal sample for further characterization.

Although not many quantitative data are available in the literature, the efficacy of the proposed methodology in providing a quantitative reduction of Ag^+^ ions is undoubted, and is likely the consequence of the very potent reducing properties of PNS [31] combined with the effectiveness of the solvent-free mechanochemical approach. In this regard, it has to be noticed that only a few examples of direct comparative studies between one-step-solid state mechanochemical synthesis and conventional wet-chemical protocols for AgNP preparation have been reported [19]. Moreover, while many reports described AgNP production by plant extracts and even bacteria and fungi related primary and secondary metabolites [11,12,13,25,52,53,54,55], relatively less have dealt with unextracted solid materials of plant origin.

### 3.2. Morphological and Structural Characterization of AgNP-PNS

ATR-FTIR analysis of AgNP-PNS (Figure 1b) indicated a significant decrease in the absorbance of the O–H bond stretching vibration band (3300 cm^−1^) compared to parent PNS, as well as of the C–H stretching of methylene (2923 cm^−1^) and of the 1615 cm^–1^ resonance of the aromatic C=C bonds [37]. These findings are in agreement with the expected involvement of the highly reducing phenolic OH groups of the B ring of the prodelphinidin PNS tannins, [32] resulting in extensive structural modifications of this latter, likely as a consequence of Ag^+^-induced oxidative dimerization leading to purpurogallin-like moieties (Figure 1).

The dimensional and morphological characterization of AgNP-PNS was performed by DLS and electron microscopy. DLS analysis probed the size and possible agglomeration of the AgNP dispersion. Figure 1c shows the intensity and number particle distributions of the sample as a function of the particle size. The intensity distribution curve showed two distinct peaks at 20 nm and 150 nm, indicating the presence of a moderate number of aggregates within the samples. Indeed, the examination of the number-based distribution confirmed that the concentration of aggregates was extremely low. On a number basis, the majority of the sample consisted of 20 nm particles, with a relative contribution of aggregates lower than 1%. This result indicates that sonication allowed to efficiently disassemble the aggregates, yielding the primary particles in the solution.

SEM and TEM observations (Figure 1d,e) of AgNP-PNS revealed the presence of small particles embedded within the organic biomass matrix. Higher magnification images indicated that the observed structures were made up of fairly spherical particles of approximately 50 nm in diameter, mostly aggregated in complex architecture clusters with dimensions ranging from 100 to 250 nm.

### 3.3. Antioxidant Properties of AgNP-PNS

Since PNS is among the most potent antioxidant agri-food by-products [31], the antioxidant properties of AgNP-PNS were evaluated as well by DPPH and FRAP assays, following the “QUENCHER” method [45]. The DPPH assay indicated a marked reduction in the antioxidant properties in comparison to starting PNS (Table 1), which was far beyond that expected based on the content of PNS in the AgNP-PNS sample (55% *w*/*w*), likely as a consequence of the efficient reduction of Ag^+^ ions by the phenolic units. Notwithstanding, the EC_50_ value exhibited by AgNP-PNS was still comparable, if not higher, to those of other phenol-rich waste materials from the agri-food sector [31,56]. Notably, less marked differences were observed between PNS and AgNP-PNS in the FRAP assay (Table 2), with both samples exhibiting very low reducing properties toward Fe^3+^ ions.

These results are in agreement with what emerged from ATR-FTIR spectroscopy analysis, i.e., a lower relative number of phenolic -OH groups in AgNP-PNS compared to PNS. Indeed, the FRAP assay determines the electron transfer capacity of an antioxidant, while the DPPH is a mixed-mode assay, as a DPPH reduction can occur both through an electron transfer and a hydrogen atom transfer mechanism [57]. Thus, the differences observed between pristine PNS and AgNP-PNS in the two assays would specifically indicate a decrease in the hydrogen atom transfer capacity of PNS, as a result of the involvement of the phenolic -OH groups in AgNP production (Figure 1). In agreement with this conclusion were also the antioxidant properties of the sample prepared using a 85:15 *w*/*w* PNS/AgNO_3_ ratio, which exhibited a significantly lower EC_50_ value (0.072 ± 0.006 mg/mL) in the DPPH assay but a comparable Trolox eqs value (43 ± 2) in the FRAP assay with respect to AgNP-PNS.

### 3.4. Photocatalytic Properties of AgNP-PNS

Several papers have reported the ability of diverse Ag^0^-doped materials to catalyze the photodegradation of organic dyes [58,59,60,61,62]. On this basis, the photocatalytic properties of AgNP-PNS were investigated by means of MB photodegradation experiments. The sample was added at a 0.4 mg/mL dose to a 30 mg/mL MB solution, which was irradiated for 120 min under visible-light generated by a solar simulator. The percentage amount of residual MB was periodically determined by UV–Vis spectroscopy, and the results are reported in Figure 2a, together with those from the control experiments.

Almost 100% of MB consumption was observed within 120 min under visible-light irradiation in the presence of AgNP-PNS, whereas only 23% dye degradation was detected in the absence of the sample. Actually, a progressive, although lower, MB consumption (up to 58%) was observed in the presence of AgNP-PNS, even in the dark. This of course could be the consequence of simple adsorption of MB on the AgNP-PNS surface. However, this kind of adsorption process has been reported to reach a plateau after ca. 30 min [60], as indeed observed when PNS was used in the place of AgNP-PNS. In addition, in the case of PNS alone, no significant differences in the dye decay were observed with and without irradiation. All these results would therefore strongly point to a catalytic, rather than an adsorption, process at the base of the MB degradation ability of AgNP-PNS. This hypothesis was confirmed by the profile of the MB degradation plot obtained from separate experiments in which the dye was left to be adsorbed on AgNP-PNS for 30 min in the dark before light exposure, showing a drop in MB amount when the solution was irradiated. In particular, the catalytic activity of AgNP-PNS could be attributed to a sustained pro-oxidant activity of the sample, also operating also in the absence of light. The reactive oxygen species generated through the well-known ability of AgNP to induce oxygen reduction [63] would be responsible for the decay of the dye observed under these conditions. The pro-oxidant activity of AgNP would obviously be further enhanced by photo-irradiation, as also observed in the case of the antibacterial activity (see Section 3.6), resulting in the more rapid consumption of MB observed under visible-light irradiation compared to the dark conditions. The role of AgNP in inducing dye photodegradation was confirmed by the lower MB consumption (82% after 120 min) detected when the sample was prepared using a 85:15 *w*/*w* PNS/AgNO_3_ ratio instead of AgNP-PNS. In another series of experiments, the recycling and stability properties of AgNP-PNS as a photocatalyst were investigated. As shown in Figure 2b, AgNP-PNS still maintained a high efficiency in the MB photodegradation experiments after three recycles. No changes in the crystal structure of AgNP-PNS before and after three-recycle measurements were apparent in the XRD patterns (Figure 2c) [64], further proving its stability.

### 3.5. Biocompatibility of AgNP-PNS

Recent advances in nanotechnology have expanded the potential applications of AgNPs in various biomedical fields, leading to a potential increase in adverse effects on human health [65]. Recent studies demonstrated that silver nanoparticles’ toxicity is mainly influenced by both the synthesis route and physico-chemical properties of the NPs [66]. Phyto-based nanoparticle synthesis overcomes the drawbacks associated with chemical routes of AgNPs synthesis, resulting in higher cell biocompatibility. Therefore, the potential cytotoxic effect of AgNP-PNS on human cells was evaluated. It was noted that the AgNP-PNS was highly biocompatible, and its toxicity was dose- and time-dependent (Figure 3), reaching a survival rate lower than 70% only at 150 μg/mL after 48 h for both cell lines (HDF and HEK).

The biocompatibility observed in both types of normal cells may be related to the use of natural compounds for AgNP synthesis, which mitigates their potential toxicity. Indeed, Khorrami et al. [67] demonstrated the selective toxicity of green synthesized silver nanoparticles against a cancerous cell line with respect to non-cancerous cells. In another study, Bin-Jumah and co-worker verified that silver nanoparticles obtained using root extracts of Beta vulgaris L [68] exerted more adverse effects on hepatic cancer cells (HuH7) with respect to normal cells (CHANG).

### 3.6. Antibacterial and Antibiofilm Activity of AgNP-PNS

The capability of AgNP-PNS in terms of inhibiting bacterial growth in dark conditions was first established by monitoring the growth rate of two pathogenic bacterial strains, namely *P. aeruginosa* PAO1 and *S. mutans*. The first is one of the most frequent agents responsible for the wound infections of soft tissue, urinary tract, bloodstream, and surgical sites, whereas *S. mutans* is involved as a primary agent causing dental caries [63]. The reported results (Figure 4) demonstrated that AgNP-PNS significantly inhibited bacterial growth already after 12 h of incubation, at concentrations ranging from 250 to 1000 μg/mL (*p* < 0.01). In particular, the most remarkable effect was observed against PAO1 at 24 h, as 54% growth inhibition (*p* < 0.001) was noted with respect to the control.

A comparable antimicrobial effect against the two bacterial strains was also observed when AgNP-PNS was analyzed in the solid medium (Figure 5). Very interestingly, a dramatically larger mean zone of inhibition was observed for AgNP-PNS at all tested concentrations when the culture discs were incubated under light irradiation. The differences in the mean zone of inhibition at all AgNP-PNS concentrations were significantly different (*p* < 0.05) against the control. It is worth noting that PNS alone also showed antimicrobial activity, however, its effect was much lower than that of AgNP-PNS.

On the basis of antibacterial assay results, and since both PAO1 and *S. mutans* have the ability to form biofilm, which is considered one of the major factors causing antibiotic resistance [63], in other experiments, the antibiofilm activity of AgNP-PNS was analyzed at different time points, at concentrations between 250 and 1000 μg/mL. Figure 6 shows a significant reduction (*p* < 0.001) in biofilm formation, as evaluated by the CV assay in the presence of 1000 μg/mL AgNP-PNS, regardless of the bacterial strain used. Moreover, already after 6 h of incubation in the presence of the sample, a reduction of about 54 ± 4% and 47 ± 2% was detected in the PAO1 and *S. mutans* biofilms, respectively. These results are in line with the reported antibiofilm activity of AgNP produced by several methodologies, and can be attributed to the ability of AgNP to alter the structure and characteristics of the biofilm by arresting the bacterial metabolic pathways involved, e.g., in motility, oxidative stress defense, respiration, and quorum sensing systems, although the precise antibiofilm as well as bactericidal mechanisms of AgNP remains partly unclear.

## 4. Conclusions

An optimized clean and scalable procedure for the synthesis of multifunctional AgNP based on PNS as an efficient and low-cost agri-food by-product is reported in the present paper. The developed protocol fulfills most of the 12 principles of Green Chemistry [69], including: (1) Prevention (no waste is produced, and a biowaste is actually recovered and reused); (2) Atom economy (PNS play both a reducing and stabilizing role and the materials used in the process are all incorporated into the final product); (3) Less hazardous chemical synthesis (PNS is a natural material); (5) Safer solvents and auxiliaries (no solvent is used); (6) Design for energy efficiency (the synthesis is performed at ambient temperature and pressure); (7) Renewable feedstocks (PNS are a largely available agri-food by-product); (8) Reduce derivatives (no nanoparticle stabilizers are needed); and (12) Inherently safer chemistry for accident prevention (non-flammable and non-toxic reagents are used; furthermore, the resulting product is biocompatible). Additionally, several principles of green engineering [70] are addressed, such as: (2) Prevention instead of treatment (no waste is produced); (3) Design for separation (AgNP-PNS is used as it is obtained, with no need for separation and/or purification); and (12) Renewable rather than depleting (agrifood byproducts are employed).

In addition, based on the criteria selected for the “greenness” evaluation of AgNP synthesis protocols [71], the proposed approach would obtain good scores on several parameters such as the reducing agent (waste, code 4), the capping agent (not needed, code 5), and the solvent (not needed, code 5).

Although a comprehensive technical and economic feasibility of AgNP-PNS manufacturing remains to be addressed, and further experiments are needed required for a complete assessment of the potential health and environmental risks associated with the use of AgNP-PNS, the reported approach is promising for the large-scale development of antimicrobial devices for healthcare, including bone tissue engineering, as well as topical antibacterial agents or wound dressing materials with antibiotic activity. Concerning the application of AgNP-PNS in catalysis, a more in-depth characterization of the stability and efficacy of the material under complex environmental conditions as well as of the photocatalytic degradation mechanism, including the aim of comparing AgNP-PNS with novel photocatalytic nanomaterials for the removal of pollutants [72,73,74], will certainly deserve attention in further investigation.

## Data Availability

Data are available on request.

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
