# Peer review of "Combining the Potent Reducing Properties of Pecan Nutshell with a Solvent-Free Mechanochemical Approach for Synthesizing High Ag0 Content-Silver Nanoparticles: An Eco-Friendly Route to an Efficient Multifunctional Photocatalytic, Antibacterial, and Antioxidant Material"

_nanomaterials, 2023, doi:10.3390/nano13050821_

Round 1

Reviewer 1 Report

The authors have described a low-cost, and simple solid-state mechanochemical protocol for the synthesis of silver nanoparticles (AgNP) based on the use of agri-food by-product shells from pecan nut  (PNS) as a reducer. Under the idealized conditions (180 min, 800 rpm, 19 PNS/AgNO3 ratio=55/45 w/w), a total reduction of silver ions was attained, culminating in a material encompassing ~ 36% w/w Ag(0).  Additionally, the ensued AgNP-PNS showed significant light-enhanced growth inhibition properties against Pseudomonas aeruginosa and Streptococcus mutans at concentrations as low as 250 μg/mL, besides eliciting an antibiofilm effect at 1000 mg/mL.

Such materials have been deployed in the past in aqueous media as shown for wine waste (pomace): High Value Products from Waste: Grape Pomace Extract - A Three-in-One Package for the Synthesis of Metal Nanoparticles.: ChemSusChem, 2, 1041-1044 (2009), which may be cited along with existing reference #24.

However, it is indeed convenient to have the solvent-free processing. The main question remains how to use this material as often solution forms are needed for deployment in any application? Further, how to determine concentration, if such dilutions were eventually carried out in presence of aassociated carbona material?

Author Response

First, we would like to thank the reviewers for their attentive and pertinent comments, which contributed significantly to improve quality and significance of our paper. Point by point replies to all Reviewers’ remarks are below. We hope that the paper is considered suitable for publication in Nanomateials.

Reviewer #1

  1. Such materials have been deployed in the past in aqueous media as shown for wine waste (pomace): High Value Products from Waste: Grape Pomace Extract - A Three-in-One Package for the Synthesis of Metal Nanoparticles.: ChemSusChem, 2, 1041-1044 (2009), which may be cited along with existing reference #24.

We thank the reviewer, the relevant paper has been included in the references.

  1. However, it is indeed convenient to have the solvent-free processing. The main question remains how to use this material as often solution forms are needed for deployment in any application? Further, how to determine concentration, if such dilutions were eventually carried out in presence of associated carbon material?

The remark of the reviewer is pertinent. We expanded the introduction including further aspects on the advantages of solvent-free processes, related to the storage, which is of particular concern in the case of NPs due to their propensity to aggregate when dispersed in a solvent (lines 49-52). As seen in the paper, the reported applications require that AgNP-PNS are dispersed in water to elicit their activity. Indeed, dispersion of AgNP-PNS in water is quite an easy task, as it requires only vortexing or sonicating the material. In particular, when sonication is applied, the biomaterial can be homogeneously dispersed due to the small particle size, and settling requires about 24 hours to occur. As for the concentration determination, we used quantitative XRD to determine the weight amount of Ag0 in the biomaterial (about 36% by weight in AgNP-PNS, see line 251), so that AgNP concentration in the liquid formulation can be controlled according to the desired dilution.

Reviewer 2 Report

This paper reports a low-cost, scalable and direct solid-state mechanochemical scheme for the synthesis of silver nanoparticles (AgNP) based on the highly reducing agricultural product Pecan shell (PNS). Adding AgNP to PNS can support the effective reduction of Ag+ ions by PNS phenolic compounds. Under visible light irradiation, AgNP-PNS can induce the degradation of methylene blue with good recovery stability. It will become a multipurpose and dual-functional material with sustainable development and easy access. Before considering publishing, there are some questions that need to be clarified. Detailed comments are as follows:

1. Why did you choose pecan to prepare AgNP? What are the advantages of this material compared with other phenolic compounds with antioxidant properties? some new Ag catalysts references should be introduced and added. Such as Molecular Catalysis, 2021, 515, 111922; Journal of Colloid and Interface Science, 2020, 571, 38-47;

2. At present, there have been many studies on agricultural food waste and other by-products of phenolic compounds with strong antioxidant properties used to prepare AgNP with antibacterial properties. What are the advantages of the synthesis methods mentioned in this paper? Please compare and explain.

3. The experiment shows that AgNP-PNS also maintains a high degradation capacity under dark conditions. Whether separate adsorption experiments have been carried out, and which is the dominant role in the degradation process of catalysis and adsorption.

4. Please provide the characterization comparison of different PNS/AgNP ratios to explain the synthesis of the materials with the best ratio, and supplement the characterization of the materials after reaction to explain its cyclicity.

5. Whether the stability of the material is tested, and whether the high efficiency can be maintained under complex environmental conditions. Whether the influence of temperature, ph, organic matter and other factors are considered in the catalytic degradation process.

6. Whether the economic feasibility of AgNP PNS manufacturing needs to be evaluated. And if the material is used in medical and health care, whether the use of AgNP PNS has potential risks and hazards to human health or the environment.

7. Some tense problems, grammatical problems and the beauty of pictures in the article need to be further improved.

Author Response

  1. Why did you choose pecan to prepare AgNP? What are the advantages of this material compared with other phenolic compounds with antioxidant properties? some new Ag catalysts references should be introduced and added. Such as Molecular Catalysis, 2021, 515, 111922; Journal of Colloid and Interface Science, 2020, 571, 38-47.

We thank the reviewer for the observation. We modified the introduction accordingly, including the suggested refences about the catalytic activity of AgNPs. Furthermore, we included more information about choosing PNS as a reducing biomass in the preparation of AgNPs at the beginning of the Results and Discussion section (lines 212-219). In this respect, preliminary experiments were carried out using PNS alongside other two very active antioxidant agrifood byproducts. As now reported in Figure S1 in the SI, PNS was selected as optimal bioreductant, since it yielded a relevant amount of Ag0 NPs without the formation of AgCl as a byproduct. We think this aspect is more clear now.

  1. At present, there have been many studies on agricultural food waste and other by-products of phenolic compounds with strong antioxidant properties used to prepare AgNP with antibacterial properties. What are the advantages of the synthesis methods mentioned in this paper? Please compare and explain.

We thank the reviewer for the observation. As already explained in the response to reviewer #1, we further detailed in the introduction section the advantages of using a solventless reaction route compared to the already reported synthesis methods. Furthermore, we recalled the advantages of the developed protocol in the conclusion sections, by a punctual reference to the principles of green chemistry and engineering. Nonetheless, a final paragraph has been also included, which summarizes the main issues still to be addressed in order to perform a comprehensive assessment of the potential health and environmental risks associated with the use of AgNP-PNS, for large scale application of this material in healthcare and catalysis.

  1. The experiment shows that AgNP-PNS also maintains a high degradation capacity under dark conditions. Whether separate adsorption experiments have been carried out, and which is the dominant role in the degradation process of catalysis and adsorption.

Almost 100% of methylene blue consumption was observed within 120 min under visible-light irradiation in the presence of AgNP-PNS, whereas only 23% dye degradation was detected in the absence of the sample. Actually, as correctly noticed by the reviewer, a progressive, although lower, methylene blue consumption (up to 58%) was observed in the presence of AgNP-PNS even in the dark. This of course could be the consequence of simple adsorption of methylene blue on the AgNP-PNS surface. However, this kind of adsorption process has been reported to reach a plateau after ca. 30 min, as indeed observed when PNS was used in the place of AgNP-PNS. In addition, in the case of PNS alone no significant differences in the dye decay were observed with and without irradiation. All these results would therefore strongly point to a catalytic, rather than an adsorption, process at the base of the methylene blue degradation ability of AgNP-PNS. This was confirmed by the profile of the methylene blue degradation plot obtained from separate experiments in which the dye was left to be adsorbed on AgNP-PNS for 30 min in the dark before light exposure, showing a drop in methylene blue amount when the solution was irradiated. The catalytic activity of AgNP-PNS could be attributed to a sustained pro-oxidant activity of the sample, operating also in the absence of light. The reactive oxygen species generated through the well-known ability of AgNP to induce oxygen reduction would be responsible for the decay of the dye observed under these conditions. The pro-oxidant activity of AgNPs would be of course further enhanced by photo-irradiation, as observed also in the case of the antibacterial activity, resulting in the more rapid consumption of methylene blue observed under visible-light irradiation compared to the dark conditions. This has been now better explained in lines 350-377 of the revised manuscript.

  1. Please provide the characterization comparison of different PNS/AgNP ratios to explain the synthesis of the materials with the best ratio, and supplement the characterization of the materials after reaction to explain its cyclicity.

We agree with the reviewer. Actually, although the optimization process of the AgNP-PNS synthesis had been undertaken, this aspect was underrated in the first submission. Therefore, we included in Section 3.1 more information on the results obtained by varying PNS/AgNO3 mass ratios (lines 222-236). Furthermore, the relevant XRD patterns have been included in the SI (Figure S2). Finally, the XRD pattern of AgNP-PNS as synthesized and after three cycles of methylene blue photodegradation has been now showed in the main text (Figure 2c).

  1. Whether the stability of the material is tested, and whether the high efficiency can be maintained under complex environmental conditions. Whether the influence of temperature, ph, organic matter and other factors are considered in the catalytic degradation process.

We agree with the reviewer that a more in-depth characterization of the performance of AgNP-PNS is needed to definitely demonstrate its possible use as a photocatalytic agent for dye degradation. However, we feel that this characterization is beyond the scope of this paper which was aimed at presenting a general overview of the multifaceted opportunities offered by our material. In any case, the limitations of our study related to this issue have been now presented in the Conclusions section (lines 481-491 of the revised manuscript.)

  1. Whether the economic feasibility of AgNP PNS manufacturing needs to be evaluated. And if the material is used in medical and health care, whether the use of AgNP PNS has potential risks and hazards to human health or the environment.

The remark raised by the reviewer is relevant, therefore cytotoxicity of AgNP PNS has been assessed on two normal human cell lines. The results are reported in Section 3.5 and Figure 3. As far as the environmental toxicity and economic feasibility are concerned, as already stated in the response to the previous remark, we also agree with the reviewer that a more comprehensive assessment of these issue should be addressed to definitely demonstrate the possible use of AgNP PNS in health and catalysis applications. However, the complete elucidation of these aspects would require a large and specifically focused effort, possibly including LCA analysis, which is outside the scope of the present work.

  1. Some tense problems, grammatical problems and the beauty of pictures in the article need to be further improved.

As can be seen from the comparison between original and revised version, the grammar and language have been completely revised, and we hope the paper is now suitable for publication.

Reviewer 3 Report

The authors developed Ag NP/PNS as an efficient multifunctional photocatalytic, antibacterial and antioxidant material. This work is interesting. However, several modifications are required before publication as follows:

1.      The standard XRD card pattern of Ag should be shown in Fig. 1A.

2.      The effect of the adding amount of PNS on the size and microstructure of the Ag NPs should be explored.

3.      The AgNP-PNS with different Ag content should be prepared and their activity should be compared.

4.      The novel photocatalytic nanomaterials for the removal of pollutants should be introduced to keep the latest research trends. e.g.: Chem. Eng. J., 2023, 455, 140943, Advanced Fiber Materials 2022, 4, 1620–1631, Sep. Purif. Technol., 2023, 304, 122401.

5.      The mechanism for the photocatalytic degradation of pollutant should be analysed and discussed in detail.

6.      Colourless pollutant should be selected as the model contaminant to evaluate its photo-activity.

7.      Adsorption-desorption tests must be performed before the start photoreaction process. The pollutant adsorption percentage on the samples must be provided and their effects on the photocatalysis need to be discussed.

8.      Why can AgNP-PNS degrade MB in dark? Please explain this in the article.

9.      The main reactive species responsible for pollutant degradation should be experimentally revealed.

Author Response

Reviewer #3

  1. The standard XRD card pattern of Ag should be shown in Fig. 1A.

We included the complete XRD pattern of Ag0 in Figure S1b, and we compared it with the XRD curve of the reaction product from mechanochemical synthesis, from which is clear that there was no contamination from AgCl.

  1. The effect of the adding amount of PNS on the size and microstructure of the Ag NPs should be explored.

As already answered to the point 4 of the Reviewer #2, we included in Section 3.1 (lines 222-236) and Figure S2 more information on the results obtained by varying PNS/AgNO3 mass ratios.

  1. The AgNP-PNS with different Ag content should be prepared and their activity should be compared.

The antioxidant and photocatalytic properties of the sample prepared using a 85:15 w/w PNS/AgNO3 ratio, containing a lowest amount of Ag(0), were tested as reported in lines 320-324 and 368-370 of the revised version, respectively.

  1. The novel photocatalytic nanomaterials for the removal of pollutants should be introduced to keep the latest research trends. e.g.: Chem. Eng. J., 2023, 455, 140943, Advanced Fiber Materials 2022, 4, 1620–1631, Sep. Purif. Technol., 2023, 304, 122401.

The suggested references have been properly quoted in the Conclusion section.

  1. The mechanism for the photocatalytic degradation of pollutant should be analysed and discussed in detail.

The proposed photocatalytic degradation mechanism has been now discussed in more details (lines 356-368 of the revised manuscript).

  1. Colourless pollutant should be selected as the model contaminant to evaluate its photo-activity.

We thank the reviewer for this insightful suggestion. Actually, we performed photodegradation experiments on naproxen under the same conditions reported in the manuscript for methylene blue, but we were not able to detect any photocatalytic effect of AgNP-PNS. However, no adsorption of naproxen on AgNP-PNS was observed by UV-vis analysis of the reaction mixture, even in the dark. This low adsorption capacity could negatively affect the photocatalytic performance of the material. Further experiments, which could be the subject of a more specialized paper, will be performed to better clarify the role of contaminant adsorption in the photocatalytic performance of AgNP-PNS as well as to expand the range of contaminants able to be photodegraded by AgNP-PNS.

  1. Adsorption-desorption tests must be performed before the start photoreaction process. The pollutant adsorption percentage on the samples must be provided and their effects on the photocatalysis need to be discussed.

Additional experiments were performed in which methylene blue was left to be adsorbed on AgNP-PNS for 30 min in the dark before light exposure. the obtained results confirmed a prominent role of photocatalysis, rather than adsorption, in the methylene blue degradation ability of the sample (see Figure 2a and lines 356-366 of the revised version).

  1. Why can AgNP-PNS degrade MB in dark? Please explain this in the article.

The ability of AgNP-PNS to degrade methylene blue in the dark could be attributed to a sustained pro-oxidant activity of the sample, operating also in the absence of light. The reactive oxygen species generated through the well-known ability of AgNP to induce oxygen reduction would be responsible for the decay of the dye observed under these conditions. The pro-oxidant activity of AgNP would be of course further enhanced by photo-irradiation, as observed also in the case of the antibacterial activity, resulting in the more rapid consumption of methylene blue observed under visible-light irradiation compared to the dark conditions. This has been now better explained also in lines 360-368 of the revised manuscript.

  1. The main reactive species responsible for pollutant degradation should be experimentally revealed.

We agree with the reviewer that identification of the main reactive species involved in methylene blue degradation by e.g. EPR experiment is required for a more in-depth characterization of the photodegradation mechanism of AgNP-PNS. However, we feel that this characterization is beyond the scope of this paper which was aimed at presenting a general overview of the multifaceted opportunities offered by our material. In any case, the limitations of our study related to this issue have been now presented in the Conclusions section (lines 481-491 of the revised manuscript).

Round 2

Reviewer 2 Report

accepted

Reviewer 3 Report

After checking the revised version, I think that the authors have well addressed the issues. In this case, this manuscript can be recommended for publication.